# Computer-aided drug design to generate a unique antibiotic family

Christopher J. Barden[1], Fan Wu[1], J. Pedro Fernandez-Murray[2], Erhu Lu[1], Shengguo Sun[1], Marcia M. Taylor[1], Annette L. Rushton[2], Jason Williams[2], Mahtab Tavasoli[2], Autumn Meek[1], Alla Siva Reddy[1], Lisa M. Doyle[1], Irina Sagamanova[1], Kovilpitchai Sivamuthuraman[1], Robert T. M. Boudreau[2], David M. Byers[3], Donald F. Weaver[1,4,5] & Christopher R. McMaster[2] ✉

The World Health Organization has identified antibiotic resistance as one of the three greatest threats to human health. The need for antibiotics is a pressing matter that requires immediate attention. Here, computer-aided drug design is used to develop a structurally unique antibiotic family targeting holo-acyl carrier protein synthase (AcpS). AcpS is a highly conserved enzyme essential for bacterial survival that catalyzes the first step in lipid synthesis. To the best of our knowledge, there are no current antibiotics targeting AcpS making this drug development program of high interest. We synthesize a library of > 700 novel compounds targeting AcpS, from which 33 inhibit bacterial growth in vitro at ≤ 2 µg/mL. We demonstrate that compounds from this class have stand-alone activity against a broad spectrum of Gram-positive organisms and synergize with colistin to enable coverage of Gram-negative species. We demonstrate efficacy against clinically relevant multi-drug resistant strains in vitro and in animal models of infection in vivo including a difficult-to-treat ischemic infection exemplified by diabetic foot ulcer infections in humans. This antibiotic family could form the basis for several multi-drug-resistant antimicrobial programs.

Antimicrobial resistance (AMR) occurs when bacteria evolve and become immune to the effects of existing antibiotics rendering them ineffective in treating infections[1–6]. A key reason behind the need for new antibiotics is the limited range of existing drugs as many antibiotics currently in use belong to a few specific classes and bacteria have developed mechanisms to evade their effects[7,8]. This narrowing of antibiotic options is leaving healthcare providers with ever-limiting choices when treating infections, increasing the risk of treatment failure[9]. Currently, it is estimated that 4.95 million deaths per year worldwide are associated with antimicrobial resistance (AMR), with 1.27 million of those directly due to AMR[2]. In essence, if there were no MDR infections at all 4.95 million lives would be saved, whereas if all

MDR infections were replaced with drug-resistant infections 1.27 million lives could be saved. The six leading causes of AMR deaths in order are *Escherichia coli*, *Staphylococcus aureus*, *Klebsiella pneumoniae*, *Streptococcus pneumoniae*, *Acinetobacter baumannii*, and *Pseudomonas aeruginosa*[2]. Of these, the single pathogen-drug combination that causes the most deaths per year is methicillin-resistant *S. aureus* (MRSA). Oddly, MRSA is listed as a high but not critical priority on the current WHO list of priority pathogens for intervention while a second species, fluoroquinolone-resistant *E. coli*, which was also a major contributor to AMR deaths, is not listed at all. AMR-associated deaths would rank third, and AMR-caused deaths twelfth, in global burden of disease deaths. Despite the above, only one direct-acting new chemical

[1]Krembil Research Institute, University Health Network, University of Toronto, Toronto, ON, Canada. [2]Department of Pharmacology, Dalhousie University, Halifax, NS, Canada. [3]Department of Biochemistry and Molecular Biology, Dalhousie University, Halifax, NS, Canada. [4]Department of Chemistry, University of Toronto, Toronto, ON, Canada. [5]Department of Pharmaceutical Sciences, University of Toronto, Toronto, ON, Canada. ✉e-mail: Christopher.mcmaster@dal.ca

entity (i.e., new antibiotic) against AMR has been approved for clinical use in the past 35 years[10–20]. Beyond the rise in AMR, the next pandemic could be bacterial. Indeed, many previous pandemics have been bacterial with the worst pandemics in human history caused by bacteria as exemplified by The Black Death (bubonic plague) pandemics due to the bacterium *Yersinia pestis*[21,22]. The evolution of bacteria is an ongoing process, constantly adapting and acquiring new ways to resist antibiotics. It is essential to stay ahead of this evolution by continually developing new antibiotics that can combat these resistant strains effectively[1,23–26].

Beyond AMR, there are current clinical settings where more effective antimicrobials are needed. A notable case is that of diabetic foot ulcer infections (DFUs). According to the International Diabetes Federation, diabetes currently affects over 540 million people worldwide and is expected to rise to 640 million by 2030. Approximately one-quarter of all diabetics will develop a DFU in their lifetime. Region-by-region prevalence of active foot ulceration is sparsely reported and highly variable, with reported prevalence up to 11% suggesting a range of 35–50 million DFUs per year[27]. The direct cost to treat diabetes in the USA is thought to be US$237 billion with US$80 billion of that for the treatment of DFUs, which compares to the estimated $80 billion required to treat cancer per year[28]. The lack of effective treatment options for DFUs results in DFUs accounting for 85% of all amputations in developed countries including 100,000 in the USA alone[28,29].

In this study, we describe a computer-aided drug design effort against the bacterial antibiotic target, AcpS, that resulted in the development of a unique, and easily synthesizable, antibiotic family of compounds that have broad efficacy against MDR Gram-positive bacteria and synergize with colistin to expand efficacy to Gram-negative bacteria. We go on to show efficacy of one member of this family in animal models of infection. Other members of this new antibiotic family can be further characterized as a starting point to treat numerous MDR infection types.

## Results and discussion

### Computer-aided drug design of AcpS inhibitors

We embarked on a de novo computer-aided drug design strategy to ensure that our potential unique antibiotic class would be structurally distinct from current antibiotics. We targeted a protein not affected by current antibiotics to decrease the odds of antibiotic resistance. This protein target is holo-acyl carrier protein synthase (AcpS), a highly conserved and essential bacterial enzyme[30–36]. AcpS is the essential starting point for lipid metabolism in bacteria via the activation of acyl carrier protein (ACP) through the donation of a 4'-phosphopantetheine moiety from coenzyme A (CoA) to produce holo-ACP (Fig. 1a)[31,37]. Our rational design of small molecules inhibiting AcpS activity was facilitated by available detailed structural knowledge of the enzyme-substrate complex and its catalytic mechanism[30–33,35,36].

Using known crystal structures of AcpS and initially evaluating multiple chemotypes to identify a molecular framework capable of geometrically precise interactions with the steric confines of the receptor pocket, we designed de novo a small, focused library to probe the generally lipophilic and cationic nature of the active site, built around a synthetically modular class of tri-substituted, penta-atomic aromatic heterocyclic compounds bearing at least one negatively charged carboxylic acid bioisostere (tetrazole) and a central ring (thiophene) that could be functionalized in a synthetically facile manner at up to three additional points. We developed this "thienyltetrazole" focused library over three generations, using docking studies to guide the selection of substituents.

The first generation of thienyltetrazoles broadly explored carboxylic acid, ester, amide, and lipophilic (mono- and bicyclic) substituents at positions 2 and 5. Within the first 50 compounds, it was established that the tetrazole needed to be "bare" and should be the only ionizable group in the molecule, as all compounds with more or less formal charge at pH 7.4 were inactive in vitro. Substituting only at 2 or 5 ablated activity, as did cyclizing to the thiophene ring; however, making dimer tool compounds (i.e., bis(thienyltetrazoles)) resulted in greatly improved potency in vitro. We arrived at an exemplar compound with pseudosymmetric aryl moieties, DNM0213, which had in vitro potency similar to that of a dimer but at a more desirable molecular weight (373.267 g/mol). We regard this compound as the culmination of the first generation (Fig. 1b).

Next, we embarked on a second generation of pseudosymmetric thienyltetrazole compounds. These maintained the 2,5-diaryl functionalization of DNM0213 while changing the phenyl substituents to incorporate a greater variety of functional group moieties intended to provide London dispersion and/or induced dipole interactions with lipophilic and/or cationic residues in the active site, respectively. A key design constraint of this generation was to incorporate a capacity for penetrance through bacterial cell walls and membranes, which is a property not readily predictable from structure. Changes to the identity or substitution pattern of the central core were also explored, and while a phenyl substituent was tolerated (Supplementary Fig. 1, including exemplar DNM0461, the highest-scored compound in docking studies), generally thiophene remained the best choice to maintain enzymatic potency while enhancing MIC. Ultimately, in the third generation we continued to optimize and balance lipophilicity with maintaining/improving enzymatic potency and MIC; both of which we were able to improve considerably over the design of the thienyltetrazole library (Fig. 1b). The "sweet spot" for this optimization is exemplar DNM0547, which was among the highest scoring compounds in docking studies (Fig. 1c).

Our compounds have all been synthesized in fewer than six steps, starting with readily available materials and avoiding difficult reaction conditions, making them amenable to scale-up for preclinical/clinical studies (for which we have developed gram-scale routes and performed larger-yield route discovery for selected leads). In total, we synthesized a library of over 700 compounds, for which we have identified significant structure-function-activity relationships against AcpS (Supplementary Fig. 1). Potency to inhibit AcpS enzyme activity improved markedly during compound development and generally corresponded to an improvement in minimum inhibitory concentration (MIC) against a broad range of Gram-positive bacteria. Using Clinical Laboratory Standards Institute (CLSI) guidelines, 33 compounds in our antibiotic family possessed a minimum inhibitory concentration (MIC) of $\leq 2 \, \mu g/mL$ against MRSA, along with an IC$_{50}$ for inhibition of AcpS enzyme activity in the 1–15 $\mu M$ range (Supplementary Table 1).

### Inhibition of AcpS by DNM0547 compounds

AcpS was chosen as a potential new antibiotic target for several important reasons beyond its high level of conservation across bacterial species. First, the $K_m$ for the substrates of AcpS is estimated to be 1.8 $\mu M$ for apo-ACP and 40 $\mu M$ for CoA[38,39]. The high $K_m$ for CoA should enable low $\mu M$ affinity compounds to effectively inhibit activity. Second, AcpS demonstrates substrate inhibition against apo-ACP and thus inhibition of AcpS activity is amplified as apo-ACP substrate level increases with prolonged inhibition of AcpS activity[38,39]. We undertook a more in-depth characterization of AcpS inhibition using DNM0547 as the test compound for this antibiotic family. The computer-aided drug design strategy used positions DNM compounds in the active site of AcpS. If DNM0547 is binding to the active site of AcpS competitive inhibition against an AcpS substrate would be predicted. We determined the activity of AcpS against increasing concentration of CoA in the absence and presence of 8 $\mu M$ DNM0547 and observed an increase in $K_m$ with no change in $V_{max}$ consistent (Supplementary Fig. 2) with competitive inhibition of AcpS by DNM0547.

Analysis of known AcpS co-crystal structures shows that residues proximate to the active site (defined as within 4.5 Å of CoA) are broadly conserved, with DNM0547 predicted to have interactions with Arg46, Phe50, and Lys63 (using *S. aureus* amino acid residue numbering). To determine if Arg46, Phe50, and Lys63 of AcpS were important for AcpS activity and DNM0547 binding, as predicted by our computer-aided drug design strategy, we used alanine scanning mutagenesis of these residues to determine their role in AcpS enzymatic activity and inhibition by DNM0547. If our predicted binding site for DNM0547 is correct then the IC$_{50}$ values should increase for the Arg46, Phe50, and Lys63 amino acid residues when mutated to Ala.

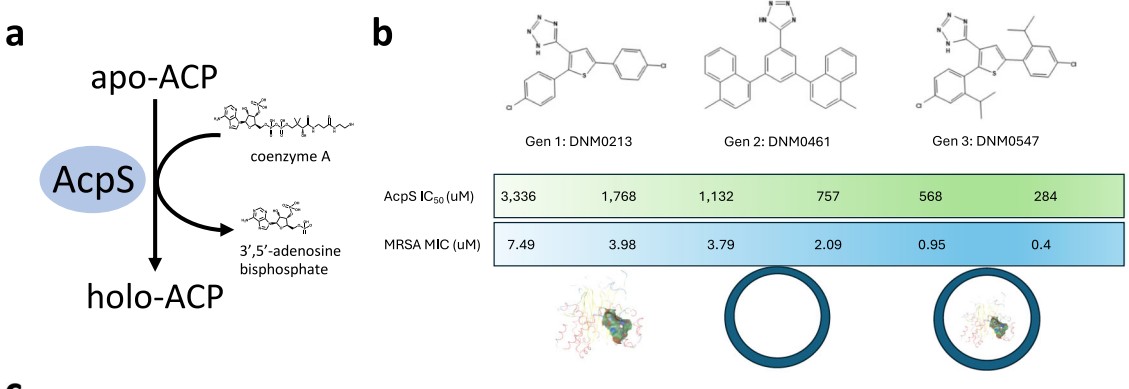

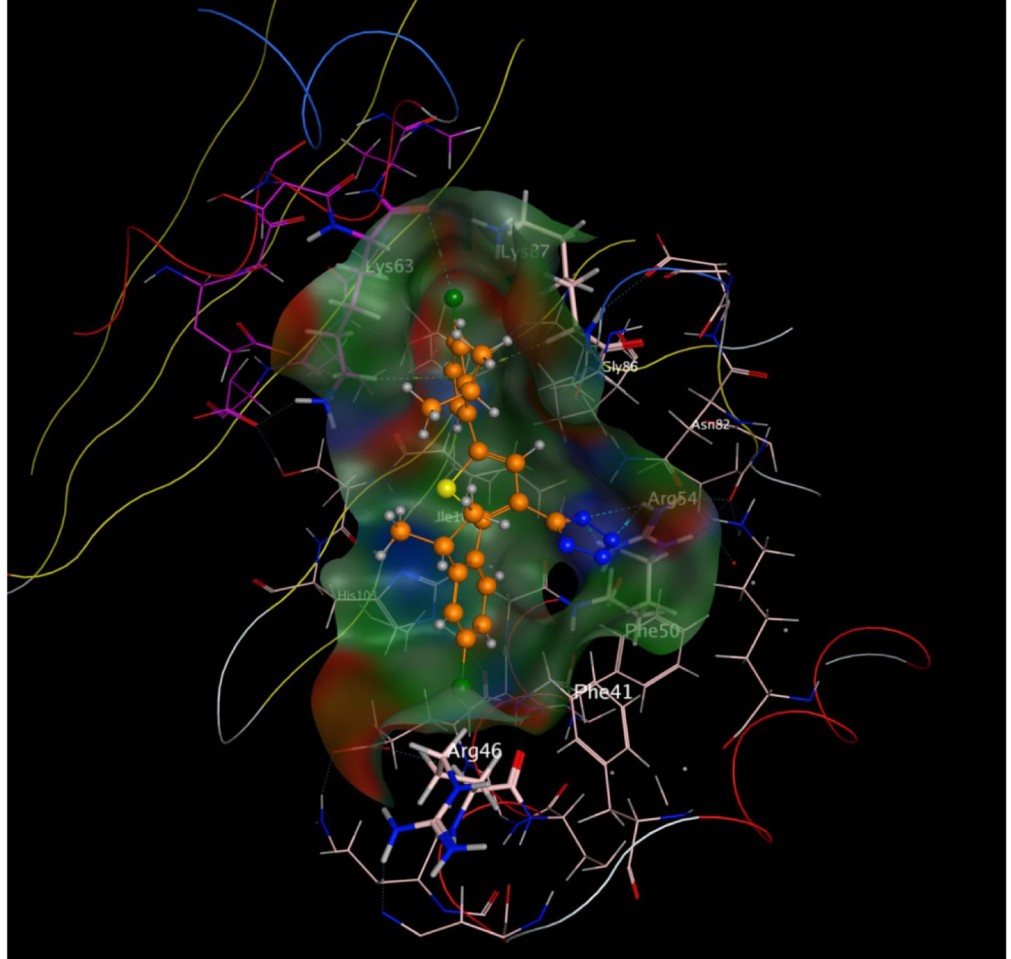

**Fig. 1 | Rationale design of AcpS active site inhibitors. a** AcpS catalyzes the transfer of a 4'-phosphopantetheine moiety from coenzyme A onto a conserved Ser residue of apo-ACP producing 3'5'-adenosine bisphosphate and holo-ACP. **b** Exemplar compounds from the three generations of thienyltetrazole-focused library. Compounds in Generation 1 (Gen 1) were optimized to find a plateau of AcpS potency and were physiochemically optimized through Generation 2 (Gen 2) for bacterial cell permeability (via MIC) as denoted by the double circle, to arrive at Generation 3 (Gen 3) compounds that can both inhibit AcpS and permeate bacterial membranes. Average values of each property are shown left to right across the library optimization. Several compounds, as the focused library evolved, rose to the potency of low μM inhibitors of AcpS IC$_{50}$ and low μg/ml MIC against the test organism, MRSA. **c** Predicted top pose (after minimization) of DNM0547 in the AcpS active site. Proximal cationic and lipophilic residues are prominently labeled: Lys63 A chain; Phe41, Arg46, Phe50, Arg54, Lys87 B chain.

Using purified *E. coli* AcpS proteins, we determined the enzyme activity of wild-type and numerous AcpS mutants. Those analyzed by scanning alanine mutagenesis included (*S. aureus* amino acid residue followed by the corresponding *E. coli* residue) R46/48, F50/50, R54/53, K63/62, N82/81, and H-/108. Enzyme activity determination of $AcpS^{F50/50A}$, $AcpS^{R54/53A}$, and $AcpS^{K63/62A}$ determined that each was inactive indicating their importance in catalysis (Supplementary Fig. 3). Interestingly $AcpS^{R46/48A}$ displayed slightly increased activity when compared to wild-type enzyme. We also tested $AcpS^{N82/81A}$ and $AcpS^{H-/108A}$ that lie within the active site but are not predicted to aid in DNM0547 binding (our modeling determines that Asn82/81 displays minimal interaction energy with DNM0547 and is 2 Å more distant than from one of the two pseudo-symmetric pi-systems of DNM0547 while His108 is over 15 Å distant and plays no appreciable role in binding) and each showed only a modest decrease in AcpS enzyme activity. The fact that the $AcpS^{F50/50A}$ mutant was enzymatically active allowed us to determine if the $IC_{50}$ against DNM0547 increased as would be predicted if $AcpS^{F50/50}$ was important for DNM0547 binding, we used the $AcpS^{N82/81A}$ and $AcpS^{H-/108A}$ as controls as they lie within the active site and were not predicted to contribute to DNM compound binding. The $IC_{50}$ for DNM0547 for wild-type AcpS was predicted to be 5.1 μM and this increased to 16.2 μM for the $AcpS^{F50/50A}$, while $IC_{50}$ values of 5.0 μM and 5.6 μM determined for $AcpS^{N82/81A}$ and $AcpS^{H-/108A}$ against DNM0547 are similar to wild type. The exploration of the proposed DNM0547 binding site by site-directed mutagenesis of expected amino acid docking residues, and competitive inhibition of AcpS enzyme activity by DNM0547, are consistent with the computer-aided prediction for DNM0547 binding to the active site of AcpS.

## In vivo testing of the DNM compounds in animal models of infection

To determine if this class of antibiotics possess in vivo antimicrobiological activity we first tested a subset of these compounds in an animal model, a rabbit ischemic ear wound model that mimics the ischemic conditions observed in DFUs[40–43]. DFUs are a current large unmet medical need for which current antibiotics are poorly efficacious and current topicals are ineffective[28,44]. Treating patients with infected wounds complicated by a limited peripheral blood supply and ischemia/hypoxia at the wound site, such as DFUs in the legs and feet of diabetes patients, is particularly difficult as the amount of systemically delivered (iv/oral) drug that can reach the infection site is dependent on a functioning vascular system. Although current topical antibiotics are ineffective against DFUs[44], topical delivery does hold many advantages in treating ischemic infections in the periphery where there is limited vascular supply[45]. First, a topical antibiotic is delivered directly to the site of infection, allowing for a higher and more sustained concentration of therapeutic agent (which has the added benefit of limiting AMR development). Second, it bypasses the need for good vascularization in the affected area for compound delivery. Third, there is a limited potential for systemic absorption and toxicity, including nephrotoxicity, allergic reactions and disruption of the body's natural bacterial flora. Fourth, topical agents are easily applied in an outpatient setting, thereby decreasing the need for institutional care. A subset of our compounds possess properties amenable to topical delivery including a high logP and small molecular weight (under 400 Da). Based on the target discovery profile, we advanced three DNM compounds, DNM0487, DNM0547, and DNM0614 into the rabbit ischemic ear wound model. In this model each compound was delivered topically for 14 days and each significantly improved wound closure; quantitative microbiology at day 14 determined that there was a > 99% reduction in bacterial survival as compared to vehicle control (Supplementary Fig. 4a and b).

DNM0547 was chosen for more in-depth study. Topical delivery of DNM0547 (2%) to the rabbit ischemic ear wound model resulted in complete wound closure by day 21 and effectively eliminated bacteria from the wound (Fig. 2a, b). Mupirocin is the leading topical antibiotic with annual sales of US$310 million, is effective against MRSA, and is currently used to treat impetigo, other skin infections, and as an MRSA de-colonizing agent. We sought to determine if DNM0547 had superior properties to the leading topical antibiotic in the treatment of ischemic wound infections. Indeed, mupirocin is not effective for the treatment of DFUs in a clinical setting and was used by us to determine the veracity of the rabbit ischemic wound model to mimic DFUs where mupirocin is ineffective. Topical delivery of mupirocin (2%) to the rabbit ischemic ear wound model did not alter wound healing from control (Fig. 2c).

We also performed an in vivo study in a cellulitis mouse model to further examine the ability of DNM0547 to treat infection. Cellulitis is an infection that occurs in the dermis and subcutaneous tissue. Mice were infected intradermally with MRSA for one day and then treated for 7 days with a cream containing DNM0547 (2%) or vehicle control. Results demonstrate that DNM0547 resulted in a 4-fold reduction in wound size and reduced bacterial load by >98% (Fig. 2d, e).

## Spectrum of DNM0547 antibacterial utility

To further determine the utility of DNM0547 in the treatment of broad spectrum Gram-positive multi-drug resistant infections we determined MICs against 102 ATCC strains and clinical isolates of *Staphylococcus*, *Streptomyces*, and *Enterococcus* and compared DNM0547 to mupirocin, vancomycin, ciprofloxacin, and erythromycin. (Supplementary Table 2). DNM0547 displayed an MIC of ≤ 2 μg/mL against all but seven of the strains tested, compared to 37 for vancomycin, 26 for mupirocin, 64 for ciprofloxacin, and 76 for erythromycin. Of the seven strains that had an MIC ≥ 2 μg/mL against DNM0547, four were 4 μg/mL, two were 8 μg/mL and one was 32 μg/mL. This demonstrates that DNM0547 can be used to effectively treat clinically relevant drug resistant strains. We also sought to determine if DNM0547 was bacteriostatic or bactericidal. Time kill experiments were carried out using increasing doses of DNM0547 against two different strains of MRSA. There was a three log drop in surviving bacteria at 2 h against 4x and 8x MIC, and at 2 h and 6 h for 2x MIC (Supplementary Fig. 5).

## AcpS is the major target of DNM0547 in vivo

Interestingly, MICs against Gram-negative bacteria were generally poor (>128 μg/mL), even though AcpS is highly conserved. The outer membrane of Gram-negative bacteria is a robust permeability barrier that prevents many antibiotics from entering Gram-negative cells[46,47]. The outer leaflet of this outer membrane is primarily composed of the unique lipid lipopolysaccharide (LPS). Strains of *E. coli* with point mutations in genes coding for LPS biosynthetic enzymes have been isolated that enable some antibiotics to effectively pass through the outer membrane enabling activity against these strains of *E. coli*[48–54]; one such strain is *E. coli* (D22) which contains a point mutation in the *lpxC* gene. The MIC of the D22 strain of *E. coli* against DNM0547 was determined to be 32 μg/mL while that for wild type (K12) *E. coli* was 256 μg/mL consistent with outer membrane permeability being a barrier to DNM0547 effectiveness against Gram-negative bacteria.

We used the fact that the D22 strain of *E. coli* did demonstrate a measurable (albeit poor) MIC toward AcpS to determine if AcpS was indeed the major drug target for DNM0547. To do so, we overexpressed AcpS in the D22 strain and determined if the increase in AcpS level resulted in a corresponding increase in MIC. Overexpression of AcpS increased the MIC of DNM0547 from 32 μg/mL to 256 μg/mL (Supplementary Fig. 6a). Over-expression of AcpS was confirmed by western blot (Supplementary Fig. 6b). This is consistent with inhibition of AcpS being the major mechanism of action for the antimicrobial activity of DNM0547.

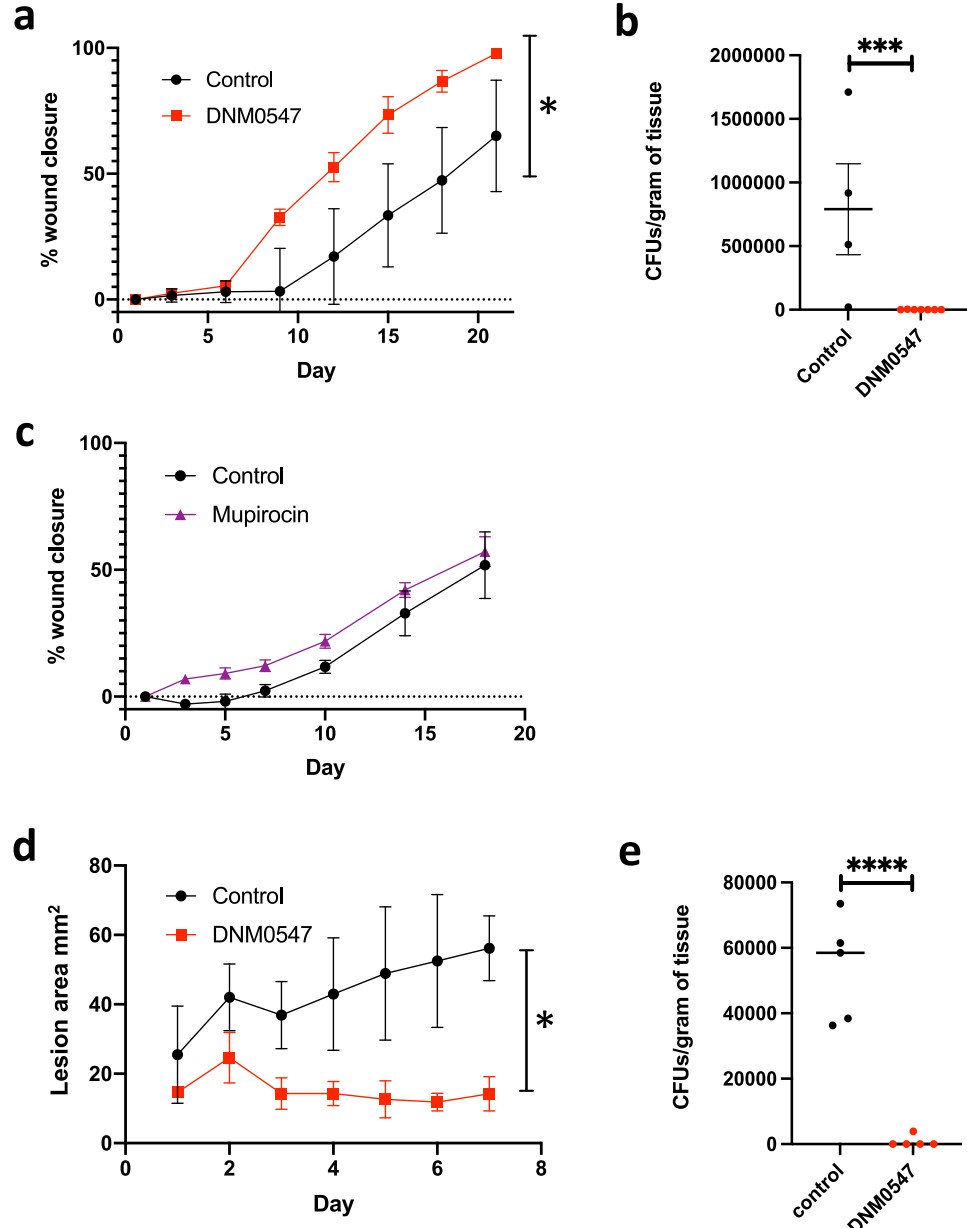

**Fig. 2 | In vivo efficacy of DNM0547. a** DNM0547 activity against MRSA (ATCC 33591) in a rabbit ischemic ear model of infection. Control contained excipient cream only while DNM0547 was 2% (w/w) in excipient; control ($n = 6$) and DNM0547 ($n = 10$). Data are represented as mean ± SEM. Significance was determined using a two-way ANOVA with Geisser-Greenhouse correction with Šidák's multiple comparisons test, $P$ 0.05. **b** CFUs per gram of tissue were determined at the end of a rabbit ischemic ear model of infection; control ($n = 4$) and 2% DNM0547 ($n = 7$). Data are represented as individual data points with the line representing the mean and error as SEM. Significance was determine using a one-sided unpaired student's $t$-test with Welch's correction, ***$P$ 0.008. **c** Mupirocin activity against MRSA (ATCC 33591) in a rabbit ischemic ear model of infection; control ($n = 6$) wounds and 2% mupirocin treated ($n = 5$). Data are represented as mean ± SEM. Significance was determined using a two-way ANOVA with Geisser-Greenhouse correction with Šidák's multiple comparisons test. Differences between control and mupirocin treated animals were not statistically significant. **d** DNM0547 activity against MRSA (ATCC 33591) in a mouse cellulitis model of infection; control ($n = 5$) and 2% DNM0547 treated ($n = 5$). Data are represented as mean ± SEM. Significance was determined using a two-way ANOVA with Geisser-Greenhouse correction with Šidák's multiple comparisons test, *$P$ 0.05. **e** CFUs per gram of tissue were determined at the end of the mouse cellulitis model of infection; control ($n = 5$) and 2% DNM0547 treated ($n = 5$). Data are represented as individual data points with the line representing the mean. Significance was determine using a one-sided unpaired student's $t$-test, ***$P$ 0.0001.

## Expanding DNM compounds to Gram-negative bacteria

We postulated that since a decrease in the ability to synthesize LPS increased the susceptibility of *E. coli* to DNM0547, some DNM compounds may result in increased susceptibility to colistin. Colistin (also known as polymyxin E) is currently one of the medications of last resort to treat multi-drug resistant Gram-negative infections[55–59]. Colistin is a polycationic peptide that preferentially binds to the LPS membrane of Gram-negative bacteria and acts by solubilizing regions of this membrane. A survey of DNM compounds determined that DNM0650, DNM0652, DNM0658, and DNM0755 substantially decreased the MIC of colistin against *K. pneumoniae*, *A. baumannii*, and *E. coli* (Supplementary Table 3). DNM0652 was chosen for further study via determination of fractional inhibitory concentration index as an indication of synergy against a panel of *Klebsiella* strains. Synergy was apparent against each strain of *Klebsiella* (Supplementary Table 4).

### Determining toxicity of DNM compounds

Subsequent to treatment of the MRSA rabbit ischemic ear infection model with topical 2% DNM0547 cream for 18 days, histopathology was performed to determine if DNM0547 was toxic to dermal cells. Wounds were excised, fixed, and routine H&E staining was performed. Wounds were examined for pathology and scored in a blinded manner. There was no indication of toxicity after 18 days of exposure of the rabbit ear wound to 2% DNM0547. All indications are consistent with increased wound healing subsequent to DNM0547 treatment as erythema, edema, angiogenesis and epithelization were improved compared to treatment with either 2% mupirocin or cream control, while granulation and inflammation were similar to control and mupirocin treatment (Supplementary Fig. 7).

A pharmacokinetic study to determine if DNM0547 (2%) could enter the circulation when applied topically showed low systemic absorption and rapid clearance. We determined that the maximum concentration present in serum was 1.02 µg/mL at 7.0 h after application with a $time_{1/2}$ of 8.7 h. By 24 h the concentration of DNM0547 had decreased to 0.06 µg/mL and by 48 h was 0.03 µg/mL (Supplementary Fig. 8).

We also determined toxicity of DNM0547 as well as DNM0652 and DNM0755 to COS7 and HEK293 cells in culture (Cell lines were obtained from ATCC). Increasing amounts of each compound were incubated with each cell line for 24 h and cell viability was determined (Supplementary Fig. 9). The lethal concentration (LC) for 50% of cell death ($LC_{50}$) for DNM0547 for COS7 and HEK293 cells was 21 µg/mL and 41 µg/mL, respectively, for DNM0652 the $LC_{50}$ was 37 µg/mL and 35 µg/mL; for DNM0755 the $LC_{50}$ was 109 µg/mL and 33 µg/mL. The estimated $LC_{100}$ for each DNM compound tested was 50 µg/mL. This favorably compares to 0.25–2 ug/mL for elimination of cell growth for susceptible bacterial strains.

In summary, a broad spectrum antibiotic class has been developed that has a unique structure and mechanism of action, is bactericidal, and is effective against antibiotic resistant strains. We determined that DNM0547 within this family is efficacious in animal models of MRSA infection and was effective against MDR strains of Gram-positive bacteria in vitro. These compounds inhibit the enzymatic activity of purified AcpS as increased expression of AcpS in bacteria results in an increase in MIC against DNM0547 demonstrating mechanism of action. DNM0547 is an effective topical antibiotic with bacterial killing properties in an important area of medical concern or which there a currently no effective solutions, ischemic infections such as those present in the extremities of DFU patients that lead to hundreds of thousands of amputations per year, or systemic spread and death if amputations are not performed in a timely manner. In addition, beyond DNM0547, there are over 30 other molecules in this antibiotic family to further explore, this new antibiotic family may form the basis for several AMR programs.

## Methods

### Computer-aided drug design and structure activity relationship

A pharmacophore model was developed by analyzing the 3D structure of the primary residues of the AcpS active sites within *B. subtilis* and *S. aureus* as given in PDB IDs 4JM7, 3F09, 1F7L. De novo design techniques were used to create candidate structures for inhibition of AcpS, and these designs were further manually refined based on biological data. The de novo design was undertaken in the sense of the International Union of Pure and Applied Chemistry (IUPAC) definition via the design of bioactive compounds by incrementation construction of a ligand model within a model of the receptor or enzyme active site, the structure of which is known from X-ray or nuclear magnetic resonance data. Using known AcpS X-ray structural data we to identify likely point of contact between putative ligand and residues near and within the AcpS active site that would bind to these residues were intuit central core structures that would combine the fragments into a synthetically feasible whole. As such, we incrementally constructed a "ligand model within a model" built from X-ray structures. Experimental data on AcpS enzyme activity inhibition and MICs informed the structure-activity relationship and refined the ligand model which was used to propose new structures, which was not an automatic process but was explicitly guided by our team of chemists to leverage the synthetic knowledge developed throughout the process, and especially to explore chemical diversity in fragments while preserving a small collection of synthetically accessible cores with reliable coupling chemistries using a manually curated, fragment-based approach. These iterative quantitative structure-activity relationships were utilized to identify important features for potency over the course of lead optimizing the >700 compounds synthesized in this program.

Specifically, docking across PDBs was performed in Molecular Operating Environment (MOE, Chemical Computing Group) originally using 3F09 PDB ID, and subsequently superseded by 4JM7 PDB ID; rankings were similar in both PDBs. Structures were prepped to fix breaks and protonation states as appropriate using the Structure Preparation tool. The Amber10:EHT force field was used, which has Amber parameters for protein atom types and EHT for small molecules. Compounds were prepped for docking by adjusting protonation to dominant state at physiological pH, calculating partial charges using AM1-BCC as expected for the force field and minimizing. The 3D region between chains bounded by A chain Lys63/Thr67, B chain His43/Arg46/Phe50, and B chain Gly86 was used as the region for docking, and in order to avoid biasing the confirmatory docking with respect to any given small molecule, the docking area was further defined as the dummy atoms placed by the Site Finder tool within that 3D region (the largest region so identified within chains A and B). Docking placement was performed using the triangle matcher algorithm (up to 1000 returned poses per molecule, retaining best 30), scored initially using London dG, then refined using reaction field GB/VI and scored using GBVI/WSA dG (9A cutoff, free side chain movement, gradient 0.01 iterations 500). Given the pseudosymmetric structure of many compounds, top poses were manually inspected to establish a consensus for the orientation within the proposed binding pocket. Final top poses shown in the Figures were further relaxed through minimization (backbones tethered).

### Minimum inhibitory concentrations

The medium employed for the assay was Mueller Hinton II Broth (MHB II; Becton-Dickinson, Sparks, MD, USA). Agar plates used for the organisms were trypticase soy agar plus 5% sheep blood (BD; Sparks, MD, USA). MIC assay plates were prepared using the CLSI broth microdilution procedure. Automated liquid handlers (Multidrop 384, Biomek 2000 and Biomek FX) were used to conduct serial dilutions and liquid transfers. A standardized inoculum of each organism was prepared per CLSI methods. Suspensions were prepared to equal a 0.5 McFarland standard, followed by a 1:20 dilution in test media. The inocula were dispensed into sterile reservoirs divided by length (Beckman Colter) and the Biomek 2000 was used to inoculate the plates. Daughter plates were placed on the Biomek 2000 work surface in reverse orientation so that inoculation took place from low to high drug concentration. The Biomek 2000 delivered 10 µL of the diluted suspension into each well. These inoculations yielded a final cell concentration in the daughter plates of ~5 × 105 colony-forming units/mL in each well. The wells of the daughter plates ultimately contained 185 µL of broth, 5 µL of drug solution, and 10 µL of bacterial inoculum. Plates were covered with a sterile lid on the top plate, placed in plastic bags, and incubated at 35 °C for ~18 h. The microplates were viewed from the bottom using a plate viewer and the MIC was read. The MIC was recorded as the lowest concentration of drug that inhibited visible growth of the organism. Uninoculated solubility control plates were also observed for evidence of drug precipitation.

For FIC testing, test ranges were set based on broth microdilution MIC test data. The wells of a standard 96-well microdilution plate (Costar) were filled with 150 μL of water in columns 2 through 12. A 300 μL aliquot of colistin at 40X the highest final concentration to be tested was added to each well in Column 1 of the plate. The Biomek 2000 was used to make eleven 2-fold serial dilutions in the "combination agent parental" plate from columns 2 through 11. The wells of the "test-agent parental" plate were filled with 150 μL of DMSO in rows B through H. Row A of this plate was filled with 300 μL of the DNM compound stock solution at 40X the highest final concentration to be tested. Serial 2-fold dilutions were then prepared from row B-G by hand using a multichannel pipette. The "daughter plates" were loaded with 180 μL of CAMHB using the Multidrop 384. The Biomek FX was used to transfer 5 μL of colistin from each well of the comparator parental plate to the corresponding well in all the daughter plates in a single step. Then a 5 μL aliquot from each well of the test agent parental plate was transferred by hand into the corresponding well of the daughter plate. Row H and Column 12 each contained serial dilutions of test agent and the combination agent alone, respectively, for determination of the MIC. This procedure was repeated for all test agents evaluated in combination with comparators. A standardized inoculum of each organism was prepared per CLSI methods. Colonies were picked from the primary plate and a suspension was prepared to equal a 0.5 McFarland turbidity standard. The suspensions were additionally diluted 1:20 in broth. A 10 μL standardized inoculum was delivered into each well using the Biomek 2000 from low to high concentration. These inoculations yielded a final cell concentration in the daughter plates of ~5 × 10⁵. Thus, the wells of the daughter plates ultimately contained 80 μL of media, 5 μL of test agent, 5 μL of the colistin, and 10 μL of inoculum. The test format resulted in the creation of an 8 × 12 checkerboard where each compound was tested alone and in combination at varying ratios of drug concentration. Plates were covered with a lid on the top plate, placed in plastic bags, and incubated at 35 °C for ~18 h. Plates were viewed from the bottom using a plate viewer. Prepared reading sheets were marked for the MIC of the combination agent, the MIC of test agent, and the wells of the growth-no growth interface for wells containing test agent and combination agent at varying ratios. The FIC was read and recorded as the lowest concentration of drug that exhibited no growth of the organism by row where agents were tested in combination. For each relevant row of the panel, the FIC index (FICI) and mean FICI was calculated by: $FIC_{drug\ A}/MIC_{drug\ A} + FIC_{drug\ B}/MIC_{drug\ B} = FIC\ index\ (FICI)$. The mean FICI for the combination was interpreted as follows: <0.5 = synergy, >0.5–4 = additive/indifferent, and >4 = antagonism. Microbiological measurements were performed and evaluated with Micromyx LLC (Kalamazoo, MI, USA).

## AcpS purification and in vitro enzyme assays

The *acpP* gene encoding ACP was amplified from *E. coli* genomic DNA and subcloned into the pGEX4T-1 vector. This construct drives the expression of glutathione S-transferase (GST) fused to the ACP open reading frame through a linker containing a consensus sequence for thrombin cleavage. The identity and veracity of *acpP* was verified by DNA sequencing. BL21 *E. coli* cells expressing GST-ACP were suspended in ice-cold phosphate buffer saline (PBS) and lysed by sonication. The lysate was made 1% Triton X-100 and rested on ice for 30 min with occasional mixing. A clear supernatant was isolated by centrifugation at 46,000 × g for 30 min, brought to 10 mM MgCl₂ and 2 mM MnCl₂ and incubated at room temperature for 1 h with occasional mixing. The supplementation of the lysate to 10 mM MgCl₂ and 2 mM MnCl₂ favors the activity of endogenous ACP phosphodiesterase (*acpH*) enriching the apo- fraction of GST-ACP. GST-ACP was affinity purified upon binding to glutathione-sepharose. The ACP moiety was eluted from the affinity resin by in-column thrombin digestion. Apo-ACP was separated from holo-ACP and acylated forms of ACP by ion-exchange chromatography on a Mono-Q 5/50 GL at pH 6.1 operated from an AKTA Pure 25 platform. The purity of the apo-ACP preparation was assessed by 20% PAGE on native gels containing 0.5 M urea.

The *acpS* gene was amplified from *S. aureus* and *E. coli* genomic DNA and inserted into a pET23b vector using standard recombinant DNA methodology such that the protein product would result in addition of a C-terminal 6x-His tag in frame with the AcpS open reading frames. The DNA sequences inserted in the vector were sequenced and found to be identical to the known *E. coli* and *S. aureus acpS* open reading frames. Recombinant His-Tagged AcpS expression was induced by IPTG addition and overnight culture. Cell lysate was obtained by sequential aspiration through 16, 18 and 21 G needles in a buffer of 50 mM Tris-HCl (pH 8.5), 10 mM MgCl₂, 2 mM DTT, 5% glycerol, and further adjusted to pH 8.1 with MES. Purification via His-Tag methodology resulted in a stock purified enzyme solution of ~1 mg/mL. In a typical AcpS enzyme activity assay, 57 μM [³H]acetyl-CoA, 2 μg apo-ACP, 10 mM MgCl₂, 5 mM DTT, 50 mM Na phosphate (pH 7.0), and AcpS in a final volume of 10 μL were incubated at room temperature in a 1.5 mL microcentrifuge tube. At 5, 10, 15, and 20 min 2 μL of reaction were added to 750 μl 10% TCA on ice. Bovine serum albumin (BSA; 20 μl of a 25 mg/mL solution) was then added to facilitate precipitation of radiolabeled protein. The samples were centrifuged at 12,000 × g for 5 min. Supernatants were removed, and the pellets were rinsed twice with 750 μl of 10% trichloroacetic acid. Pellets were resuspended in 50 μl of formic acid and transferred to scintillation vials. 2 mL of scintillation fluid was added, and the amount of [³H]-labeled holo-ACP formed was quantified by liquid scintillation counting. The enzymatic reaction proceeded in a linear manner depending on time (10–40 min) and enzyme concentration (1–5 nM). Routinely, each data point was determined at least by triplicate.

## Rabbit model of ischemic infection by MRSA

New Zealand White rabbits from Charles River were employed for this study. Both male and female rabbits were used; sample size prevented disaggregation of data by sex. Investigators were blinded to group allocation, data collection, and data analysis. Food and water were given ad libidum and animals were acclimatized for 7 days with photoperiods of alternating 12 h light and dark at 21 °C in relative humidity of 40–50%. The formulation for delivery of DNM compounds was Glaxal base (WellSpring Pharmaceutical Canada Corp., Oakville, Canada). Formulation was performed by dissolving compound in Glaxal base to give final concentrations of 1% or 2% DNM compound (w/v). The analgesic (Buprenex 0.02 mg/kg sc) and anesthesia (isoflurane (0.5–5% with oxygen carrier 0.5–2 L/min) were used as required during procedures. An ischemia wound healing model was conducted surgically. Each wound was inoculated with 50 μl of 2 × 10⁷ CFU/mL MRSA (ATCC #33591) and dressed. After 1 day, wounds were treated with either 50 μl control cream containing vehicle or 50 μl cream containing compound daily. Wounds were assessed and observations recorded using the Draize scoring system. Digital photographs of each wound were obtained and documented.

Post-wounding, under anesthesia, animals were euthanized with an intravenous injection of Beuthanasia-D at a dosage of 150 mg/kg of body weight. Specimens including the entire wound, along with ~2–3 mm of the normal unwounded skin margins were harvested. Each ear wound specimen was sectioned into halves. For one half of the wound, the wound was placed into a container with 10% neutral buffered formalin and processed for routine histological evaluation. Wounds were scored on a scale of 0–4 at the wound site for erythema, edema, inflammation, granulation, angiogenesis, and epithelialization. Wounds were characterized in a blinded manner by a board-certified member of the College of American Pathologists. The other half of the wound was weighed in a sterile Petri dish, then placed in a sterile disposable tissue grinder and homogenized for quantitative microbiology. The tube containing the ground tissue sample was filled with

5 mL 0.85% sterile saline. Five 1:10 serial dilutions were then prepared with 0.5 mL aliquots and 4.5 mL of sterile saline per aliquot. Subsequently, 0.1 mL of the homogenate and each dilution was drop plated on trypticase soy agar and placed in an incubator at 37 °C. The number of viable organisms per wound was recorded as cfu/gram of tissue. The rabbit ischemic ear model was performed and evaluated with Pluris Research Inc (Franklin, TN, USA). Pluris Research Inc has Public Health Service (USA) policy on humane care and use of animals ethics approval (D21-01111).

### Mouse cellulitis model of infection
SKH1 mice from Charles River were used and male and female mice were utilized, sample size prevented disaggregation of data by sex. Mice were housed with a 12 h light and dark cycle at 21 °C in relative humidity of 40–50%. Investigators were blinded to group allocation, data collection, and data analysis. The formulation for delivery of DNM0547 was Glaxal base (WellSpring Pharmaceutical Canada Corp., Oakville, Canada). Formulation was performed by dissolving compound in Glaxal base to give final concentrations of 2% DNM0547 compound (w/v). MRSA (ATCC #33591) were grown to mid-log phase, concentrated by centrifugation and suspended in sterile PBS at a stock concentration of $2 \times 10^{11}$ CFU/mL.

Four week old mice were anesthetized and injected intradermally with $1 \times 10^8$ CFU/mouse. 4 h after injection, mice were treated with 50 μl 2% DNM0547 cream, or excipient alone. Lesion dimensions were recorded daily for seven days. Food and water were given ad libidum. On day seven, mice were euthanized and the lesion area aseptically removed to 3 mL sterile PBS. The samples were homogenized and serial dilutions were plated on blood agar and incubated at 37 °C to determine CFU. The mouse model of skin infection was performed and evaluated with Washington Biotechnology Inc (Baltimore, MD, USA). Washington Biotechnology Inc has Public Health Service (USA) policy on humane care and use of animals ethics approval (D16-00616).

### Cell culture toxicity
Stock solutions of the test agents were prepared in DMSO (40 mg/mL) and stored at −20 °C until use. AlamarBlue HS™ cell viability reagent was acquired from Invitrogen (Cat # A50100, Carlsbad, CA). COS7 and HEK293 cells (obtained from ATCC) were cultured and tested at 37 °C in a 5% $CO_2$/100% humidity environment in DMEM media supplemented with 10% fetal bovine serum. Cells were trypsinized, centrifuged, resuspended in cell media, counted on a hemacytometer, and diluted to an appropriate cell density to give the desired number of cells per 300 μl media. Cells were then loaded into 48-well cell culture plates (3548; Corning, Inc.); 300 μl/well at 60,000 cells per well. The cell density was determined in preliminary tests to give linear AlamarBlue HS™ readings over a 3.5 h period for HEK293 cells and 2 h period for COS7 cells after 2 days of growth; 2 wells per plate were loaded with 300 μl/well of cell-free media to provide background readings. Plates were cultured overnight, after which test agent solutions were prepared in media at the highest test concentration (100 ug/mL) and at five serial dilutions, each containing 0.25% DMSO and 0.125% Kolliphor® EL (Cat# 27963, Sigma Aldrich). The compounds concentrations were then loaded in separate columns via 300 μl/well. Vehicle-only (0.25% DMSO and 0.125% Kolliphor® EL) medium was added to the two remaining columns on each plate to provide control viability readings. The plates were incubated for a 24 h period after which 30 μl of AlamarBlue reagent was added to each well; the plates were incubated another 3.5 h, and absorbance was read at 580 nm and 600 nm. Normalized values representing the amount of AlamarBlue converted to its oxidized form were calculated for each well using background-corrected absorbances as described by the Alamar-Blue HS protocol information packet available from Invitrogen Technical Support. The fraction of control viability for each agent concentration

with each cell line was then determined by the ratio of normalized values for the treated wells to the control wells. Estimates were then made of the agent concentration yielding 50% reduction in cell viability, labeled $LC_{50}$. Values were determined by linear interpolation of fraction of control growth against concentration for the test agent concentrations.

### Statistics and reproducibility
No data were excluded from the analyses, all experiments were randomized, blinding was used for all animal experiments, and experiments were highly reproducible. Sample sizes were chosen based on previous work to determine efficacy for interventional studies in rabbit and mouse models[60–63]. Statistical analyses were performed using Graphpad Prism version 10.1.2.

### Reporting summary
Further information on research design is available in the Nature Portfolio Reporting Summary linked to this article.

## Data availability
Source data are provided with this paper. Data supporting the findings of this manuscript are also available from the authors upon request. Source data are provided with this paper.

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

## Acknowledgements

Research was funded by Canadian Institutes for Health Research grants to DFW and CRM (SOP-159230).

## Author contributions

C.J.B.: conceptualization, formal analysis, methodology, project administration, visualization, supervision, writing—review, and editing. F.W.: conceptualization, formal analysis, methodology, validation. J.P.F.-M.: formal analysis, methodology, validation. E.L.: formal analysis, methodology. S.S.: formal analysis, methodology. M.M.T.: formal analysis, methodology. A.L.R.: formal analysis, methodology. J.W.: formal analysis, methodology. M.T.: formal analysis, methodology. A.M.: formal analysis, methodology. A.S.R.: formal analysis, methodology. L.M.D.: formal analysis, methodology. I.S.: formal analysis, methodology. K.S.: formal analysis, methodology. R.T.M.B.: formal analysis, methodology. D.M.B.: conceptualization, formal analysis, methodology, project administration, funding acquisition. D.F.W.: conceptualization, formal analysis, methodology, project administration, funding acquisition, project administration, visualization, supervision, writing—review and editing. C.R.M.: conceptualization, formal analysis, methodology, project administration, funding acquisition, project administration, visualization, supervision, writing—original draft, writing—review, and editing.

## Competing interests

A university spin-out company, DeNovaMed Inc, has been assigned patents related to parts of the work described. C.J.B., F.W., E.L., M.M.T., A.L.R., D.F.W., and C.R.M. have shares in DeNovaMed Inc. The remaining authors declare no competing interests.
