## [Peer Review File · Nature Communications]

Computer-aided drug design to generate a unique antibiotic familyREVIEWER COMMENTS

Reviewer #1 (Remarks to the Author):

The presented paper contains some interesting information about the development of a novel class of antibiotics that is urgently required due to the insurgence of drug resistance strains from several pathological microorganism. Accordingly, the idea is interesting.

However, there are some relevant issues that preclude the publication in high rank journal like this in this current form. Below are reported some points that should be necessarily addressed for reevaluating the paper.

-Introduction section should be improved with the aim of the work. In the current version there is only a background regarding bacteria drug resistance but no info about the approach to overcome this issue.

-Results and discussion

-it is not clear the rationale for the optimization of the molecule. How the authors find the reference structure for the optimization? Furthermore, authors synthesized >700 derivatives without any rationale (i.e. for improving PK parameters). After that some of them based on docking were evaluated against the selected enzyme. AT row 65 authors reported Using known crystal structures of AcpS and computer modeling we designed a receptor model. What is the sense of designing a receptor model? considering that the authors used a crystal structure of the receptor?

-Computational part methods. There is no information about de novo design that considering that authors synthesized a large amount of compounds it is not possible to affirm that this is de novo design. Information about docking parameters are missed (i.e. software, energy grid, etc). Furthermore, it is not reported the thresholds for selecting compounds for the biological evaluation

-what is the rationale for considering DNM0487, DNM0547, and DNM0614 for further studies and DNM0547 for the in vivo studies considering its weak inhibitory potency against the enzyme? why the authors did not consider DNM0753 that showed a significant potency against the enzyme?

-considering that for the tested compounds with few exception the IC50 is weak in the micromolar range, and considering that the MIC is quite good, in my opinion it is possible that the selected compounds inhibit the bacteria growth also by targeting other enzymes in

addition to the selected one, demonstrating not unique mechanism of action.

-no number of independent experiments were reported in particular for the IC50 calculation. Usually for the IC50 there is an error that should be considered accordingly SEM should be introduced in the table and the dose response curves added to the supplementary file.

-Authors reported a toxicity studies on two cell lines. Considering that they performed in vivo test, it could be better to consider also the toxicity in vivo.

-Considering that authors performed in vivo test a preliminary pharmacokinetics profile of the compound DNM0547 is highly recommended to be inserted in the paper.

-1H and 13C spectra for the final compounds should be provided in the supplementary file

-Figure 1 C and D should be revised since the picture of the docking is very bad as well as the superposition of the enzymes in which it is not possible to capture any information.

Reviewer #2 (Remarks to the Author):

The work presented in the manuscript describes a new antibiotic class with a unique structure versus holo-acyl carrier protein synthase (AcpS) using computer aided drug design. Relevant activity studies demonstrate that compounds from this class have broad Gram positive activity including efficacy against clinically relevant multi-drug resistant strains in vitro and in animal models of infection in vivo. In addition, DNM0547 increased expression of AcpS in bacteria results in an increase in MIC versus DNM0547 demonstrating mechanism of action. However, the authors have only indirectly proved that this class of compounds act on AcpS, without further verifying whether they act directly on AcpS at the molecular level. Therefore, the authors should systematically study the mechanism of this class at the molecular level.

Reviewer #3 (Remarks to the Author):

This is a topic of great importance and it is inspiring to see creative expert science applied to the global challenge of AMR

The drug target and design concept have potential to be clinically relevant and are novel,

with constructive focus on topical delivery. Perhaps the challenges of topical antibiotic use and encouraging resistance should be touched upon?

There were some areas where more detail would be helpful from a clinical perspective, some of which which also relate to reproducibility

1. The animal models are not stated as diabetic
2. The number of animals/wounds is not stated until reading the legends p.25, which means data interpretation/significance is obfuscated, especially as numbers are low
3. Using mupirocin as a control is relevant re activity against MRSA. However, statements about mupirocin imply that it is a useful agent in clinical practice (lines 137-8). It was not made clear that while mupirocin is used topically for MRSA e.g nasal carriage, it is not an evidence-based approach as used topically for diabetic foot infections.
4. When the term clinically relevant drug resistant strains is used, it should state pertains to MRSA only e.g line 152
5. When referring to broad spectrum coverage e.g conclusion line 209, the animal models were MRSA only. The MICs on cultures were tested with gram-ve and +ve bacterial strains, so should clarify in vitro, but instead says 'efficacious in animal models' as a broad spectrum entity?
6. The volume of cream applied to the rabbits was specified (line 336), but no volume was stated for the mouse cellulitis model?

I do not have sufficient expertise in all the techniques used to feedback here regarding whether all the methods are sound

Response to reviewers' comments

Reviewer 1

The presented paper contains some interesting information about the development of a novel class of antibiotics that is urgently required due to the insurgence of drug resistance strains from several pathological microorganism. Accordingly, the idea is interesting. However, there are some relevant issues that preclude the publication in high rank journal like this in this current form. Below are reported some points that should be necessarily addressed for reevaluating the paper.

1. Introduction section should be improved with the aim of the work. In the current version there is only a background regarding bacteria drug resistance but no info about the approach to overcome this issue.

Thank you for pointing to the need for an Introduction that expanded the rational for work including an expansion of the study rationale. The introduction has almost doubled in size to both point to the urgent need for new antibiotics as well as the aims of the study (p.4) Specifically, for the project aims “In this study, we describe a computer-aided drug design effort versus a novel bacterial antibiotic target, AcpS, that resulted in the development of a novel, and easily synthesizable, antibiotic family of compounds that have broad efficacy versus MDR Gram-positive bacteria and synergize with colistin to expand efficacy to Gram-negative bacteria. We go one to show efficacy of one member of this family in animal models of infection. Other members of this new antibiotic family can be further characterized as a starting point to treat numerous MDR infection types.” In essence, we hope the community will be encouraged to test other members of this antibiotic family via PK, distribution, and antibiotic properties in vitro and in vivo – with this study providing evidence that DNM0547 is effective in animal models of infection and in vitro versus a plethora of MDR bacteria.

2. It is not clear the rationale for the optimization of the molecule. How the authors find the reference structure for the optimization? Furthermore, authors synthesized >700 derivatives without any rationale (i.e. for improving PK parameters). After that some of them based on docking were evaluated against the selected enzyme. AT row 65 authors reported Using known crystal structures of AcpS and computer modeling we designed a receptor model. What is the sense of designing a receptor model? considering that the authors used a crystal structure of the receptor?

The rationale for optimizing the molecules has been expanded in the Result and Discussion section (pp. 5-6). We clarify the amino acid residues (Arg48, Phe49, and Lys62 side chains) within the pharmacophore became the main basis for compound binding/optimization and added almost half a page of text to further clarify the computer-aided drug design process. In addition to further address the computer-aided pharmacophore several additional experiments were undertaken. For these, we used DNM0547 and demonstrated competitive inhibition as would be expected of an active site inhibitor (new Suppl Fig 2). Second, we used site-directed scanning alanine mutagenesis

for the proposed DNM compound class binding residues (Arg48, Phe49, and Lys62) and the increase in IC50 for DNM0547 binding was consistent with this region being the pharmacophore (new Suppl Fig 3).

3. Computational part methods. There is no information about de novo design that considering that authors synthesized a large amount of compounds it is not possible to affirm that this is de novo design. Information about docking parameters are missed (i.e. software, energy grid, etc). Furthermore, it is not reported the thresholds for selecting compounds for the biological evaluation

An expanded section can be found in the Methods section (pp.14-15) describing the de novo design process is now included in the manuscript

4. What is the rationale for considering DNM0487, DNM0547, and DNM0614 for further studies and DNM0547 for the in vivo studies considering its weak inhibitory potency against the enzyme? why the authors did not consider DNM0753 that showed a significant potency against the enzyme?

As stated in our newly improved Introduction, the rationale for this work was to show that this new family of compounds targeting AcpS can be used as a starting point to treat numerous MDR infection types. We originally focused on DNM0487, DNM0547, and DNM0614 as at that point we had hit a floor as far as improving AcpS enzyme inhibition and MICs versus Gram-positive bacteria. We made the decision at that time to move into animal models (our capacity to design/synthesize molecules is approximately one per week in our academic setting). As all three molecules tested were efficacious in the rabbit ischemic ear model for MSRA infection, and our further characterization of DNM0547 showed efficacy in other models of infection as well as broad specificity in vitro versus MDR bacteria, this demonstrates that many other molecules in this family are worth further characterization by the broader community (including DNM0753).

5. Considering that for the tested compounds with few exception the IC50 is weak in the micromolar range, and considering that the MIC is quite good, in my opinion it is possible that the selected compounds inhibit the bacteria growth also by targeting other enzymes in addition to the selected one, demonstrating not unique mechanism of action.

Thank you for pointing out that the choice for AcpS and its inhibition required further clarification. AcpS was chosen as a novel antibacterial target for several important reasons beyond its high level of conservation across bacterial species. These are now pointed out in the revised paper (pp. 7-8). First, the K_m for the substrates of AcpS are estimated to be 1.8 μM for apo-ACP and 40 μM for CoA. The high K_m for CoA should enable low μM affinity compounds to effectively inhibit activity. Second, AcpS demonstrates severe substrate inhibition versus apo-ACP, and thus inhibition of AcpS activity is amplified as apo-ACP substrate level increases with prolonged inhibition of AcpS activity. Thus, as AcpS is inhibited there is a feed forward loop that increases apo-ACP an inhibition of AcpS activity. Also, to demonstrate AcpS is the target in vivo we over-expressed AcpS in bacteria and this effectively increased the MIC versus DNM0547 – which as pointed out by Reviewer 2 demonstrates mechanism of action (Suppl. Fig 7).

6. No number of independent experiments were reported in particular for the IC₅₀ calculation. Usually for the IC₅₀ there is an error that should be considered accordingly SEM should be introduced in the table and the dose response curves added to the supplementary file.

Each IC₅₀ and MIC, when exploring the broad class of the over 700 DNM compounds as they were being designed/synthesized, was performed three times singly. This is now indicated in Suppl. Table 1. As part of our in depth characterization of the AcpS pharmacophore we show new data on the dose response curve with error bars for a more in depth study of DNM0547 inhibition and IC₅₀ determination versus AcpS (Suppl Figs 2 and 3), along with various site-directed mutants in the proposed DNM0547 binding site that were requested by Reviewer 2 as part of a more in depth exploration of the pharmacophore. With this new data generated for the in depth study of DNM0547, we did not to show each AcpS IC₅₀ curve for the remaining 32 compounds as this will take up a lot of journal space (and we are already at 10 Suppl Figs and 4 Suppl Tables).

7. Authors reported a toxicity studies on two cell lines. Considering that they performed in vivo test, it could be better to consider also the toxicity in vivo.

To assess toxicity, as DNM0547 was delivered topically in vivo skin toxicity was examined and there was no evidence of toxicity. Indeed, increased skin healing was observed (Suppl Fig. 8). Maximum tolerated dose experiments are not allowed at our animal care facility (death cannot be an endpoint).

8. Considering that authors performed in vivo test a preliminary pharmacokinetics profile of the compound DNM0547 is highly recommended to be inserted in the paper.

As requested, we performed PK for DNM0547 (2%) delivered topically in a rodent model and determined plasma levels over time (Suppl Fig 5). The maximum concentration present in serum was 1.02 µg/ml at 7.0 hrs after application with a time_{1/2} of 8.7 hrs. By 24 hrs the concentration of DNM0547 had decreased to 0.06 µg/ml and by 48 hrs was 0.03 µg/ml (Suppl Fig 5). DNM0547 is an effective topical antibiotic with minimal systemic exposure subsequent to topical application.

9. ¹H and ¹³C spectra for the final compounds should be provided in the supplementary file

As requested, the NMR spectra have been supplied for DNM0547 as an exemplar (Suppl Fig 10). Supplying NMR data for all 33 compounds would take up a lot of journal space. As per journal policy, NMR spectra will be provided upon request for any of the other compounds.

10. Figure 1 C and D should be revised since the picture of the docking is very bad as well as the superposition of the enzymes in which it is not possible to capture any information.

As requested, Fig 1 C and D have been replaced and the figure legend updated.

Reviewer 2

The work presented in the manuscript describes a new antibiotic class with a unique structure

versus holo-acyl carrier protein synthase (AcpS) using computer aided drug design. Relevant activity studies demonstrate that compounds from this class have broad Gram positive activity including efficacy against clinically relevant multi-drug resistant strains in vitro and in animal models of infection in vivo. In addition, DNM0547 increased expression of AcpS in bacteria results in an increase in MIC versus DNM0547 demonstrating mechanism of action. However, the authors have only indirectly proved that this class of compounds act on AcpS, without further verifying whether they act directly on AcpS at the molecular level. Therefore, the authors should systematically study the mechanism of this class at the molecular level.

1. As requested, to verify how AcpS acts at the molecule level several lines of investigation were undertaken (pp 7-8). Using DNM0547 as the exemplar for this class (as it was the compound most studied in animal models in this study) we first demonstrated that DNM0547 results in competitive inhibition of AcpS enzyme activity, as would be expected of an active site inhibitor (Suppl Fig 2). Second, we used site-directed scanning alanine mutagenesis for the proposed DNM compound class binding residues (Arg48, Phe49, and Lys62) as defined by our computer-aided model and determined that “To determine if Arg48, Phe49, and Lys62 AcpS were important for AcpS activity as predicted by our computer-aided drug design strategy, we used alanine scanning mutagenesis of these residues to determine their role in AcpS enzymatic activity and inhibition by DNM0547. If our predicted binding site for DNM0547 is correct then the IC₅₀ values should increase for the Arg48, Phe49, and Lys62 residues mutated to Ala. Using purified proteins, we determined that the enzyme activity AcpS^{F49A}, AcpS^{R53A} and AcpS^{K62A} were inactive indicating their importance in catalysis (Suppl. Fig 3). Interestingly AcpS^{R48A} displayed slightly increased activity versus wild type enzyme. We also tested AcpS^{N81A} and AcpS^{H108A} that lie within the active site but were not predicted to aid in DNM0547 binding (our modeling determines that Asn81 displays minimal interaction energy with DNM0547 and is 2 angstrom more distant than from one of the two pseudosymmetric pi-systems of DNM0547 while His108 is over 15 angstroms distant and plays no appreciable role in binding) and each showed a moderate decrease in AcpS enzyme activity. The fact that the AcpS^{F49} mutant was enzymatically active allowed us to determine if the IC₅₀ versus DNM0547 increased as would be predicted if AcpS^{F49} was important for DNM0547 binding, we used the AcpS^{N81A} and AcpS^{H108A} as controls as they lie within the active site and were not predicted to contribute to DNM compound binding. The IC₅₀ for DNM0547 for wild type AcpS was predicted to be 5.1 μM and this increased to 16.2 μM for the AcpS^{F49}, while IC₅₀ values of 5.0 μM and 5.6 μM were determined for AcpS^{N81A} and AcpS^{H108A}, respectively. The AcpS enzyme activity competitive inhibition by DNM0547, and exploration of the proposed pharmacophore by site-directed mutagenesis of expected amino acid docking residues, are consistent with the computer-aided prediction for DNM0547 binding to AcpS.” Albeit a lot of work, we thank the review for stating this point as the study is now much more robust.

Reviewer 3

This is a topic of great importance and it is inspiring to see creative expert science applied to the global challenge of AMR. The drug target and design concept have potential to be clinically relevant and are novel, with constructive focus on topical delivery. Perhaps the challenges of topical antibiotic use and encouraging resistance should be touched upon?

The challenges/opportunities of why topical use can be important are now included throughout the manuscript (e.g. in infections in areas where there is poor vascularization).

1. The animal models are not stated as diabetic.

The rabbit ischemic ear model is meant to mimic the hypoxia/ischemia present in extremities of diabetics. The rabbits do not have diabetes, however, this is the best model to mimic ischemic wounds that we are aware of. This is now made clear in the revised manuscript (pp. 9-10). Thank you for asking for this clarification.

2. The number of animals/wounds is not stated until reading the legends p.25, which means data interpretation/significance is obfuscated, especially as numbers are low.

The number of animals per experiment is now clear in Figure legend 2.

3. Using mupirocin as a control is relevant re activity against MRSA. However, statements about mupirocin imply that it is a useful agent in clinical practice (lines 137-8). It was not made clear that while mupirocin is used topically for MRSA e.g nasal carriage, it is not an evidence-based approach as used topically for diabetic foot infections.

We agree that mupirocin is used clinically versus MRSA topically for infections such as impetigo and nasal carriage and this is now stated in the manuscript (p. 10) We also now state that mupirocin was used as a comparator for the rabbit ischemic wound model as there are no effective topical treatments or DFU. Since mupirocin is the most prescribed topical antibiotic, it was used as a control to show that veracity of the rabbit ischemic wound model and the capacity of DNM0547 to effectively cure these wound while mupirocin was ineffective.

4. When the term clinically relevant drug resistant strains is used, it should state pertains to MRSA only e.g line 152.

Throughout the manuscript, we now clarify that DNM0547 was effective versus a clinically relevant drug strain of MRSA for animal models, and that many MDR strains were tested via MICs for in vitro studies.

5. When referring to broad spectrum coverage e.g conclusion line 209, the animal models were MRSA only. The MICs on cultures were tested with gram-ve and +ve bacterial strains, so should clarify in vitro, but instead says 'efficacious in animal models' as a broad spectrum entity?

The wording is now “We determined that DNM0547 within this family is efficacious in animal models of MRSA infection and was effective against essentially all MDR strains of Gram-positive bacteria in vitro.”

6. The volume of cream applied to the rabbits was specified (line 336), but no volume was stated for the mouse cellulitis model?

We now state that 50 ul of DNM0547 (25) cream was used in the mouse model (p. 22).

REVIEWER COMMENTS

Reviewer #1 (Remarks to the Author):

Despite the attempts to address my previous concerns, some points are not addressed:

-the screening part continues to be not clear. Authors did not clarify, what is the designed a receptor model.

-Authors used the crystal structures, it is quite strange considering that the PDB ID 3F09 is obsolete and no longer available from PDB. After that the numbering is wrong according to the PDB file and UniprotKB: Authors wrote Arg48, Phe49, and, Lys62. This numbering does not correspond to the sequence. For 4JM7 in pos 48 there is a Ile (Arg is 46), in position 49 there is a Glu (Phe is 50) and in position 62 there is a Ser while the Lys is 63. Concerning the 1F7L, in pos 48 there is a GLu, Arg is 45, in pos 49 and 62 ok there are a Phe and IYs, respectively. Now, authors also indicated the trimeric arrangement of receptor (assembly only available for 4JM7). So the question is what receptor was used for docking calculations? considering that the numbering reported was wrong for each of both proteins available, also making a trimeric form starting from the monomer 1F7L. Materials and methods section does not help to understand this relevant issue about the computer-aided approach.

Fig 1, docking picture continue to be difficult to understand. Moreover, in the picture there are no a pharmacophore model, but only, presumably a compound with some residues without label with different representation. What information should be taken from that picture. In fact, authors claimed to have developed a pharmacophore model but there isn't, maybe the authors have not clear what is the meaning of pharmacophore model and how should be correctly developed. Accordingly, the screening procedure continue to be not clear. The early selection is not clear how was carried out (threshold for docking score is not reported and/or fitness against a potential pharmacophore that is not present.

Finally, authors wrote in the materials and methods...optimization of the receptor structure in the presence of each compound...what is the protocol for optimization? software? in the materials and methos for in silico part there is no any indication for reproducing the computational protocol (software, procedures and parameters).

Accordingly, without a rigorous workflow, explanation and understandable pictures concerning computer-based part, screening and selection of compounds, I cannot support

the publication of an article in high-rank journal without a clear rationale.

Reviewer #2 (Remarks to the Author):

The authors have conducted several investigations and verified how DNM0547 acts on AcpS at the molecular level. The results obtained are satisfactory. I recommend publication after supplementing HRMS data for DNM0547.

Reviewer #4 (Remarks to the Author):

Reviewer 3 responses

The revised manuscript describes a considerable body of work in support of AcpS inhibitors as novel antimicrobials. These include in-vitro investigation of a lead compound identified using computer-aided drug design, against a large collection of Gram-positive MDR clinical isolates. In-vivo testing in two animal wound models indicate low toxicity and improved wound healing compared to untreated controls. Synergistic gram-negative activity with colistin was also found.

In relation to the authors responses to comments of reviewer 3.

0. The authors have now included a reasonable explanation of why topical antimicrobials have potential for clinical application, particularly in the setting of diabetic foot infections, on page 9. For the manuscript to flow well, some of these details, e.g. the global picture, could be moved to the introduction, which currently contains no reference to topical antimicrobials or the challenge of DFIs and contains some superfluous details. Another advantage of topicals, not mentioned, is the potential to limit AMR development.

It is also important to cite current recommendations and rationale (topicals not currently recommended based on poor evidence) from the most recent guidelines for treatment of DFI (Senneville É, Albalawi Z, van Asten SA, Abbas ZG, Allison G, Aragón-Sánchez J, Embil JM, Lavery LA, Alhasan M, Oz O, Uçkay I, Urbančič-Rovan V, Xu ZR, Peters EJG. IWGDF/IDSA guidelines on the diagnosis and treatment of diabetes-related foot infections (IWGDF/IDSA 2023). *Diabetes Metab Res Rev.* 2024 Mar;40(3):e3687. doi: 10.1002/dmrr.3687. Epub 2023

Oct 1. PMID: 37779323).

1. The animal wound models used are not diabetic and the authors have now stated that the rabbit ischaemic wound model serves as a mimic of DFU/DFI. However, use of the model is not well explained. Is this an accepted mimic? some reference to the literature might be useful. Rather, a detailed high-level introduction to diabetes and DFU is given as the rationale. A cellulitis mouse model was also used. This section should be re-organised to state the two models used, earlier in the section. The section also includes in-vitro work which is not aligned well to the section title on page 9.

2. Numbers of animals used – authors have specified this in Figure 2 and other figures. Numbers remain missing from Supplemental Fig 8 legend.

3. Use of mupirocin.

This comment has been adequately addressed. However, line 206 should also state that it is used as a MRSA decolonising agent in addition to treating infection.

4. Specifying MRSA rather than clinically relevant drug resistant strains.

This has been addressed throughout. Please note however, line 36, reference to 'nightmare bacteria' ...this description refers to one group, carbapenamase-producing enterobacterales and is somewhat misleading here.

5. Spectrum of activity in-vitro versus in-vivo - addressed.

6. Volume of cream used – addressed.

REVIEWER COMMENTS

Reviewer #1

Despite the attempts to address my previous concerns, some points are not addressed: -the screening part continues to be not clear. Authors did not clarify, what is the designed a receptor model. Authors used the crystal structures, it is quite strange considering that the PDB ID 3F09 is obsolete and no longer available from PDB. After that the numbering is wrong according to the PDB file and UniprotKB: Authors wrote Arg48, Phe49, and, Lys62. This numbering does not correspond to the sequence. For 4JM7 in pos 48 there is a Ile (Arg is 46), in position 49 there is a Glu (Phe is 50) and in position 62 there is a Ser while the Lys is 63. Concerning the 1F7L, in pos 48 there is a Glu, Arg is 45, in pos 49 and 62 ok there are a Phe and Lys, respectively. Now, authors also indicated the trimeric arrangement of receptor (assembly only available for 4JM7). So the question is what receptor was used for docking calculations? considering that the numbering reported was wrong for each of both proteins available, also making a trimeric form starting from the monomer 1F7L. Materials and methods section does not help to understand this relevant issue about the computer-aided approach.

Thank you for pointing this out. The original PDB file (PDB ID 3F09) was used early on at the outset of this focused small molecule library design which began in 2010 as it was current at that time. As the project proceeded, newer AcpS structures became available. Docking was routinely performed as AcpS structures emerged, with PDB ID 4JM7 superceding PDB ID 3F09 in 2013 as the project moved forward. With respect to amino acid numbering, thank you for noticing the discrepancies. These have been corrected throughout.

Fig 1, docking picture continue to be difficult to understand. Moreover, in the picture there are no a pharmacophore model, but only, presumably a compound with some residues without label with different representation. What information should be taken from that picture. In fact, authors claimed to have developed a pharmacophore model but there isn't, maybe the authors have not clear what is the meaning of pharmacophore model and how should be correctly developed. Accordingly, the screening procedure continue to be not clear. The early selection is not clear how was carried out (threshold for docking score is not reported and/or fitness against a potential pharmacophore that is not present).

We feel the docking picture is now clear. DNM0547 docked into the AcpS active site is now depicted in Fig 1c with the Figure legend text and Results text denoting the requisite amino acids predicted to bind DNM0547 (which were then shown to be correct via the site-directed mutagenesis experiments that follow in the manuscript). We have also

significantly rewritten and expanded the Results section on how the focused library of compounds was developed. In essence, two assays were performed to determine compound efficacy, AcpS enzyme activity inhibition and MICs vs MRSA. Computer-aided drug design was useful for improving AcpS enzyme activity, however, there are no predictive algorithms for entry of new compounds into bacterial cells, so this process was a little more trial and error from a compound development and design standpoint. Fig. 1b seeks to illustrate this as the first generation of DNM compounds were mostly optimized based on inhibition of AcpS enzyme activity to a point where MICs did not further improve, with the second generation of DNM compounds seeking to improve MICs (which was much less computer-aided and much more trial and error and rationalization), followed by a third generation of DNM compound optimization that again used computer aided drug design to optimize compounds from the second generation. This resulted in a focused compound library of 33 compounds that can effectively inhibit AcpS and have MICs in the range of known antibiotics. From this list, the manuscript goes on to characterize DNM0547 in animal models of infection. Of course, all compounds and/or their routes of synthesis will be made available upon request for others to work on to determine their capacity as potential antibiotics.

Finally, authors wrote in the materials and methods...optimization of the receptor structure in the presence of each compound...what is the protocol for optimization? software? in the materials and methods for in silico part there is no any indication for reproducing the computational protocol (software, procedures and parameters).

The Methods section has now been expanded as requested and now describes the software and processes used during the development of the focused library of AcpS inhibitors in detail.

Accordingly, without a rigorous workflow, explanation and understandable pictures concerning computer-based part, screening and selection of compounds, I cannot support the publication of an article in high-rank journal without a clear rationale.

We feel that the processes used to develop our focused library are now very clear from the extended now provided in information in the Results and Methods sections. As with many drug development programs, part of the process was computer-aided while other parts were intuitive and/or based on the expertise of the organic and medicinal chemists as far as feasibility of compound synthesis and development and MIC optimization. A major point is that we did get to an important result – the development of a focused library of easily synthesizable compounds with antibacterial activity in vitro and in animal models.

Reviewer #2

The authors have conducted several investigations and verified how DNM0547 acts on AcpS at the molecular level. The results obtained are satisfactory. I recommend publication after supplementing HRMS data for DNM0547.

Thank you for recommending publication. The high resolution mass spec (HRMS) data is now presented alongside the NMR data for DNM0547 in the Supplementary Data section as Supplementary Fig. 10c and d.

Reviewer #4 (former Reviewer 3)

The revised manuscript describes a considerable body of work in support of AcpS inhibitors as novel antimicrobials. These include in-vitro investigation of a lead compound identified using computer-aided drug design, against a large collection of Gram-positive MDR clinical isolates. In-vivo testing in two animal wound models indicate low toxicity and improved wound healing compared to untreated controls. Synergistic gram-negative activity with colistin was also found.

0. The authors have now included a reasonable explanation of why topical antimicrobials have potential for clinical application, particularly in the setting of diabetic foot infections, on page 9. For the manuscript to flow well, some of these details, e.g. the global picture, could be moved to the introduction, which currently contains no reference to topical antimicrobials or the challenge of DFIs and contains some superfluous details. Another advantage of topicals, not mentioned, is the potential to limit AMR development. It is also important to cite current recommendations and rationale (topicals not currently recommended based on poor evidence) from the most recent guidelines for treatment of DFI (Senneville É, Albalawi Z, van Asten SA, Abbas ZG, Allison G, Aragón-Sánchez J, Embil JM, Lavery LA, Alhasan M, Oz O, Uçkay I, Urbančič-Rovan V, Xu ZR, Peters EJG. IWGDF/IDSA guidelines on the diagnosis and treatment of diabetes-related foot infections (IWGDF/IDSA 2023). *Diabetes Metab Res Rev.* 2024 Mar;40(3):e3687. doi: 10.1002/dmrr.3687. Epub 2023 Oct 1. PMID: 37779323).

Thank you for these suggestions. We have moved the global picture on topical antimicrobials and their potential for clinical application, particularly in the setting of diabetic foot ulcer infections, to the Introduction along with the current recommendation that topicals are not recommended due to poor evidence of current topicals (the above suggested DFI reference is now included in that section). In addition, we have added to the Results section an additional advantage of topicals, which is the potential to limit AMR

development, as requested.

1. The animal wound models used are not diabetic and the authors have now stated that the rabbit ischaemic wound model serves as a mimic of DFU/DFI. However, use of the model is not well explained. Is this an accepted mimic? some reference to the literature might be useful. Rather, a detailed high-level introduction to diabetes and DFU is given as the rationale. A cellulitis mouse model was also used. This section should be re-organised to state the two models used, earlier in the section. The section also includes in-vitro work which is not aligned well to the section title on page 9.

We have added references for the rabbit ischemic ear model as an accepted mimic for ischemic wounds. This model has been in use for 21 years and is very well established – the references include three recent reviews outlining its usefulness and limitations. As requested, we also reorganized the text to state the two models used earlier in the Results section and repositioned the in vitro work to a separate (following) section.

2. Numbers of animals used – authors have specified this in Figure 2 and other figures. Numbers remain missing from Supplemental Fig 8 legend.

The number of animals used is now included in the Supplemental Fig 8 legend. Four animals/wounds per treatment were analyzed.

3. Use of mupirocin. This comment has been adequately addressed. However, line 206 should also state that it is used as a MRSA decolonising agent in addition to treating infection.

The use of mupirocin as a MRSA de-colonizing agent is now included where requested.

4. Specifying MRSA rather than clinically relevant drug resistant strains. This has been addressed throughout. Please note however, line 36, reference to ‘nightmare bacteria’ ...this description refers to one group, carbapenamase-producing enterobacteriales and is somewhat misleading here.

We removed this line and reference.

5. Spectrum of activity in-vitro versus in-vivo - addressed.

6. Volume of cream used – addressed.

REVIEWERS' COMMENTS

Reviewer #1 (Remarks to the Author):

Authors addressed the concerns on the computer-based part. I have no further comments

Reviewer #4 (Remarks to the Author):

The authors have addressed all of comments in response to my initial review and they are satisfactory. I have no further concerns in relation to edits requested.

Response to reviewers' comments

All reviewers are in favour of publishing this study. There are no further comments to address from the reviewers.